# How Does the Assessment of Work Organization during the COVID-19 Pandemic Relate to Changes in the Well-Being of Health System Workers?

**DOI:** 10.3390/ijerph18158202

**Published:** 2021-08-03

**Authors:** Loreta Kubilienė, Aušra Griciūtė, Vilma Miglinė, Milda Kukulskienė, Aurima Stankūnienė, Nida Žemaitienė

**Affiliations:** 1Faculty of Public Health, Lithuanian University of Health Sciences, 44307 Kaunas, Lithuania; ausra.griciute@lsmuni.lt (A.G.); milda.kukulskiene@lsmuni.lt (M.K.); nida.zemaitiene@lsmuni.lt (N.Ž.); 2Community Well-Being Center, Mykolas Romeris University, 08303 Vilnius, Lithuania; vilma.migline@mruni.eu; 3Department of Drug Technology and Social Pharmacy, Faculty of Pharmacy, Lithuanian University of Health Sciences, 44307 Kaunas, Lithuania; aurima.stankuniene@lsmuni.lt

**Keywords:** COVID-19 pandemic, mental health, organizational factors, survey, well-being of healthcare workers

## Abstract

In the case of various emergencies, especially pandemics, healthcare workers are faced with disproportionate pressures. Organizational support plays a significant role in protecting the psychological and physical health of healthcare workers. This interdisciplinary research aims to determine how changes in the physical and psychological well-being of healthcare and pharmacy workers during the first wave of the COVID-19 lockdown are related to work organization factors that support safety and stability. A quantitative research strategy was applied in the research. Data from an electronic survey assessed the changes in the physical and psychological well-being of healthcare and pharmacy workers during the lockdown period and the organizational factors supporting safety and stability. The sample of the quantitative research consisted of 967 employees of healthcare institutions and pharmacies in Lithuania. This research broadens the concept of organizational factors and provides data on their interaction with the changes of employee well-being indicators in a pandemic situation. It was found that positive changes in the evaluation of physical as well as psychological well-being during the COVID-19 lockdown could be consistently predicted by all the analyzed safety and stability supporting organizational factors that were found to be associated with subjective physical well-being and psychological well-being even when adjusting for the effect of socio-demographic factors (gender, age, work field, and specialty). The identification and proper management of organizational factors was significant for the psychological and physical well-being of healthcare workers during the lockdown period. It was found that all estimates of safety and stability supporting organizational factors during the first wave of the COVID-19 pandemic lockdown were positively related and could act as protective factors to the subjective physical and psychological well-being of healthcare and pharmacy workers.

## 1. Introduction

The COVID-19 pandemic requested quick and effective organizational solutions: decisions taken at both the state and internal organizational levels correlated with the effectiveness of pandemic management [1,2]. A very important aspect of managing the pandemic caused changes in work organization change strategy. Consequently, it is very important to learn a lesson, have a vision, pay attention to the preparation, and implement a change plan strategy, because clear preventive steps and preparation for various situations contribute to the effective functioning of employees in the organization. The main aspects of crisis management are a clear optimistic vision, a realistic action plan, decisive action, open, honest, and frequent communication with employees, and acknowledgement [2].

During this pandemic, health system workers survived uncertainty, experienced confusion and fluster in work order, and intense changes in work function, responsibilities, and workload. During these changes in the organizations there was a lack of balance, structure, and clarity; accordingly, health care workers encountered big challenges, including conditions of increased stress and a burnout syndrome [3,4,5]. Meanwhile, clear solutions and the systematic management of health threats positively affected employee functioning in the work environment. The action plan and organization support are indicated as the main components of sustainability [2,3,4]. In response to the avalanche of change, emergency management and active leadership are considered significant aids, and the protection of the physical and mental health of workers is singled out as a priority: health care specialists want to be sure that their organization will support them and their families both medically and socially [2].

Researchers have referred to the significant role of the organization for employees during the COVID-19 pandemic: support of the organization, caring for its employees, and promoting information, leadership, and mutual assistance have been singled out as organizational factors that help the pandemic challenges to be overcome [2,3,4].

During the COVID-19 pandemic, healthcare and pharmacy workers not only faced an increased workload, they also experienced severe emotional overloads that could harm both their physical and mental health [4,5]. Hamouche (2020) outlined the four main stressors over the pandemic: safety threat, information overflow, perception of captivity and social isolation, and work-related losses [6]. If staff feel insecure, it can cause stress and burnout symptoms and reduce the ability to work effectively [7]. These symptoms can develop into post-traumatic stress disorder or other chronic diseases [2]. In relation to coping styles, it was found that a positive attitude of specialists in the workplace, when individuals positively reinterpret negative situations, is one of the main protective factors against distress [8] and it relates to self-efficacy, greater psychological well-being, and a better quality of life [9].

The regular and appropriate provision of information, psychosocial support, and ensuring physical security are important organizational factors that support personal safety and stability during emergencies [10]. This corresponds to the principles of first aid during emergencies. However, there are important protective components for psychological well-being during a pandemic. These include, but are not limited to, organizational support and appropriate training [3]. Organizational support plays a significant role in protecting the psychological and physical health of healthcare workers [3]. Perceived organizational support is defined as the employee’s perception that the organization cares about his or her well-being in a physical and psychological sense and is concerned with the employee’s commitment to work and the organization [11,12]. Favorable working conditions in the organization and the equal distribution of resources have a positive effect on the employee’s well-being and position in the organization. When employees feel supported by their organization, leaders, and environment, they become more involved in the work and master tasks better [12]. It is important that leaders also understand and recognize the needs of healthcare specialists [2,10,13].

An employee who feels good and safe in the organization can complete a better-quality job and be more attentive. The organizational climate that stimulates the psychological resilience of employees correlates with their assessment of the organization and solidarity with its values [7,12,13]. This interdisciplinary research aims to determine how changes in the physical and psychological well-being of healthcare and pharmacy specialists during the first wave of COVID-19 lockdown are related to work organization factors that support safety and stability. It was hypothesized that healthcare and pharmacy workers with a better assessment of physical and psychological well-being during the first lockdown would tend to assess work organizational factors better. Therefore, it is very important to understand the problem in order to suggest well-balanced solutions which would help the organizations reduce the risks related to the physical and emotional health of their employees.

## 2. Materials and Methods

### 2.1. Procedure

The article presents research data taken from the Research Council of Lithuania funded project “COVID-19 Pandemic-related Challenges, Psychological Well-being and Support Needs of Healthcare Workers and Pharmacists” (No. P-COV-20-44), the aim of which was to assess the physical and psychological well-being of healthcare workers and pharmacy specialists and the challenges of work organization and assistance needs related to the pandemic caused by the coronavirus COVID-19. This article analyzes part of the data collected during the project related to the aspects of work organization and employees’ well-being. Quantitative data on the coping strategies of the employee and subjective media exposure, as well as qualitative data were also collected, which, due to the extensive scope of the data, are not analyzed in this publication.

To allow employees of healthcare institutions and pharmacies throughout Lithuania to safely participate in the research, it was conducted remotely. Respondents were invited to answer the questions of the submitted electronic questionnaire on the Lithuanian University of Health Sciences data collection platform specially programmed for this research. Unique links to the electronic questionnaire were generated, which, together with the invitation to participate in the research, were sent via institutional e-mails to 222 target healthcare institutions and pharmacy chains. A convenient sampling strategy was applied. Healthcare institutions had been selected on the basis of a Minister of Health order, which designated coordinating and service delivery healthcare institutions in the regions of the country during COVID-19. Data collection took place from 17 August to 15 October in 2020 after the first lockdown from 16 March to 16 June. Employees from 56 institutions throughout Lithuania voluntarily responded to this invitation and participated in the research. All pharmacy chains were included.

#### Ethics

The research was conducted with the consent of the Kaunas Regional Biomedical Research Ethics Committee (No. BE-2-88; 20 July 2020). The respondents were presented with informed consent (and contacted for more information), which they confirmed remotely before completing the questionnaire. Strict confidentiality and anonymity requirements were observed throughout the research.

### 2.2. Participants

The research sample consisted of 967 employees of healthcare institutions and pharmacies in Lithuania—857 (88.6%) women and 101 (10.4%) men. Lithuanian healthcare and pharmacy workers took part in the research: physicians, nurses, pharmacy specialists, administrative staff, and other health system workers (psychologists, technical and cleaning staff, etc.).

### 2.3. Safety and Stability Supporting Organizational Factors

To reveal how healthcare and pharmacy workers evaluated safety and stability supporting organizational factors during a pandemic (SAS-SOF) in the institutions they work, the researchers developed a separate questionnaire. Based on the experience of the researchers and the scientific literature, a larger number of statements were initially formulated to assess the supporting organizational factors [7,11,13]. Empirical measures of COVID-19 organizational support and its associations with healthcare workers’ physical health and well-being were examined [6,7,11]. The statements formulated in the course of the work were adapted and modified to allow the most accurate assessment of the respondents’ opinion on the safety and stability supporting work organizational factors in the context of COVID-19. In our research, an adapted framework was formulated to link to the Shanafelt et al. (2020) named framework of five areas to work organizational environment (Table 1) [13]. The final SAS-SOF items group included 8 statements—every SAS-SOF item was assigned to one of the specific organizational support fields according to the adapted framework and the Shanafelt et al. (2020) framework.

Before the research, the SAS-SOF questionnaire was submitted for a pilot peer review (*N* = 20) by healthcare and pharmacy workers to determine their understanding of the formulated statements and to determine the duration of the research. Several statements in the questionnaire were adjusted to integrate the observations of the respondents in the preliminary research.

Respondents were asked to rate the extent to which they agreed or disagreed with each of eight statements using a 5-point Likert scale from ‘strongly disagree’ (=1) to ‘strongly agree’ (=5). Further calculations included the respondents whose number of answers to each statement of the SAS-SOF group were at least 70% of all possible answers [14]. Missing values (1 answer was not given by 23 respondents, 2 answers were not given by 4 respondents) were included using the mean estimate of each study participant (Person mean imputation procedure).

The dependence of SAS-SOF statements on the same group was confirmed by factor analysis indicators (Kaiser, Meyer and Olkin KMO = 0.917; Bartlett‘s test of sphericity χ^2^ = 4,968,528, df = 28, *p* < 0.001 (Principal Axis Factoring analysis was applied)) and internal compatibility indicators (Cronbach’s Alpha 0.923). The statements of the SAS-SOF group explained 60.34% of the variance. The weights of separate SAS-SOF statements ranged from 0.848 to 0.726. The correlations between the SAS-SOF items were 0.777–0.485 (*p* < 0.001). The internal consistency of the analyzed responses to the SAS-SOF statements was also confirmed by the split-half reliability coefficients: Cronbach’s Alpha part 1 (4 items) 0.830; part 2 (4 items) 0.870; Spearman-Brown 0.937; Guttman split-half 0.937.

When analyzing the SAS-SOF *indicators* in the research, i.e., responses to individual statements and the total estimate—higher scores indicate better estimates in the area under analysis. The total SAS-SOF indicator was calculated by summing the estimates of eight SAS-SOF statements (range 8–40 points).

#### 2.3.1. Physical and Psychological Well-Being

The respondents were asked to rate changes in physical and psychological well-being during the period from 17 August 2020 to 15 October 2020. The research compared the distributions of the SAS-SOF indicators in the categories of assessment of physical well-being and psychological well-being: 1. decreased, 2. increased/has not changed.

#### 2.3.2. Socio-Demographic Characteristics

The research analyzed socio-demographic indicators: gender, age, workplace (public, private), position (physician, nurse, pharmacy specialist, administrative staff, and other staff).

### 2.4. Data Analysis

Statistical calculations were performed by IBM SPSS Statistics for Windows, Version 25.0 (IBM Corp, Armonk, NY, USA). Descriptive statistics and characteristics of the SAS-SOF indicators and the socio-demographic characteristics were calculated: distributions by analyzed groups in units (N) and percentages (%); mean estimates and standard deviations (SD). The affiliation of the analyzed SAS-SOF statements to the same group was assessed using factor analysis, internal consistency (Cronbach Alpha) criteria, and Split-half reliability coefficients. The comparison of the distributions of the analyzed indicators with the normal distribution was performed with the help of the Shapiro–Wilk criterion. It was found that the distributions of SAS-SOF indicators according to physical and psychological well-being groups did not correspond to the normal distribution, therefore the non-parametric Mann–Whitney U test (calculated indicators: *p* (significance level), mean ranks, Z value, Mann–Whitney U) was used to compare the distributions of these indicators by groups (*N* ≥ 30 and the Levene test confirmed the hypothesis of variance equality). When the hypothesis of variance equality was not confirmed, the Chi-square test was applied. Binary logistic regression was used to assess the associations between SAS-SOF indicators and the respondents’ changes in physical well-being and psychological well-being during the lockdown (1. decreased, 2. increased/has not changed). In order to assess the impact of the differences in a person’s socio-demographic characteristics on the relationship between SAS-SOF indicators and the changes in respondents’ well-being indicators, a multivariate binary logistic regression was applied when controlling for independent variables: gender, age, workplace, profession. Logistic regression results are presented: p, OR (probabilities ratio) with 95% confidence intervals [15]. The level of significance was set at *p* < 0.05.

## 3. Results

The characteristics of the research participants’ distributions according to the groups of socio-demographic data and physical and psychological well-being categories are presented in Table 2. We can see that a higher percentage of women (89.4%) than men (10.6%) provided the survey data—a similar ratio of gender percentage was observed in the groups of physical and psychological well-being. Two and a half times more respondents were working in public institutions (71.6%) than those working in private institutions (28.4%). A similar percentage of physicians (27.1%), nurses (26.7%), and pharmacy specialists (24.5%) represented a sample of respondents by occupation. The smallest number of the respondents were administrative staff (8.0%) and respondents of various other professions in the field of healthcare. The mean age of the respondents was 42.64 (SD = 12.72).

The characteristics of the SAS-SOF indicators are presented in Table 3. The mean overall estimate of the SAS-SOF group according to the results of the research was 25.89 points (SD = 8.89), the mean estimates of individual SAS-SOF statements range from 2.72 to 3.72 points (SD ranges from 1.27 to 1.49).

The distributions of the SAS-SOF indicators by physical well-being groups and psychological well-being groups (from the beginning of the lockdown to now 1. decreased, 2. increased/has not changed) were compared. It was found that healthcare and pharmacy workers who indicated that their physical well-being increased or had not changed from the beginning of the lockdown until the research were more likely to score higher on each statement in the SAS-SOF group (Mann–Whitney U = 49,063–60,735, *p* < 0.001), compared to the respondents who reported that their physical well-being decreased (Table 3). Respondents who indicated that their psychological well-being increased or had not changed from the beginning of the lockdown until the research were more likely to score higher on each statement in the SAS-SOF group (Mann–Whitney U = 69,799–81,416, *p* < 0.001), compared to the respondents who reported that their physical well-being decreased (Table 3).

Healthcare and pharmacy workers who confirmed that their physical well-being and/or psychological well-being increased or had not changed from the beginning of the lockdown until the research were more likely to have higher overall SAS-SOF estimates (physical well-being: Mann–Whitney U = 50,909, *p* < 0.001; psychological well-being: Mann–Whitney U = 70,783, *p* < 0.001), compared with the respondents who reported a decreased physical well-being and/or psychological well-being (Table 3).

Data from the univariate analysis established (Table 4) that the increase in the SAS-SOF estimates of the indicators was associated with an increased possibility (OR) that respondents would rate their physical well-being during the lockdown as improved or unchanged (individual SAS-SOF statements OR range from 1.23 to 1.56; total SAS-SOF OR = 1.06). The research data also showed that the increase in the SAS-SOF estimates of the indicators was associated with an increased possibility that respondents would rate their psychological well-being during the lockdown as improved or unchanged (individual SAS-SOF statements OR range from 1.23 to 1.45; total SAS-SOF OR = 1.05).

A multivariate binary logistic regression analysis was used to control the effect of independent variables: gender (women, men), age, workplace (public, private), and profession (physician, nurse, pharmacist, administrative staff, other professions) (Table 4). The results of the research confirmed the positive associations between all the researched SAS-SOF indicators and the subjectively assessed increase in physical well-being of healthcare workers and pharmacy specialists (individual SAS-SOF statements OR range from 1.23 to 1.56; total SAS-SOF score OR = 1.07) and psychological well-being (individual SAS-SOF statements OR ranges from 1.25 to 1.49; total SAS-SOF score OR = 1.06).

## 4. Discussion

During the first lockdown from 16 March to 16 June in 2020, the healthcare and pharmacy workers who participated in the research mentioned that they had felt changes in their physical and psychological well-being: 23% (*N* = 212) of the research participants stated that their physical well-being had decreased since the beginning of the lockdown and even 38% (*N* = 351) of the research participants reported a decrease in psychological well-being. There is evidence in the literature that healthcare and pharmacy specialists are negatively affected by a pandemic, both physically and psychologically [16]. Therefore, at the state, organizational, and individual levels, it is necessary to look for ways to reduce the risks associated with the physical and emotional health of workers during extreme situations.

Analyses of the scientific literature showed that we can presume that organizational support could contribute to employee well-being. According to the research data, we found that healthcare and pharmacy workers with a better assessment of physical and psychological well-being during the first lockdown tended to better assess each of the analyzed safety and stability supporting organizational indicators in the work environment. The research data gained estimates the positive association between organizational support during extreme situations and the well-being of healthcare workers and highlights the critical role of the organization in supporting the well-being of healthcare professionals and other essential workers. There are still quite intensive debates in the literature related to the critical factors for such organizational support [17]. We will discuss the safety and stability supporting organizational factors during extreme situations that were included in our research. It was found that all of the selected aspects of work organization allowed the assessment of physical and psychological well-being to be predicted in a way that was statistically significantly within certain limits.

In our research, we attributed the organization’s concern for the psychological needs of employees (attention to the psychological climate of the organization and the additional concern expressed for the health and well-being of employees) to these factors. The research data showed that the organizational climate (2.97) and additional benefits (2.88) were rated worse than the average. The literature emphasizes that concern for the organization’s climate and the safety and well-being of employees encourages employees to show solidarity with the organization and its values and helps employees to perform their work better [2]. Long-term research has highlighted the positive relationship between work-related psychosocial factors (such as effort-reward imbalance, perceived organizational support and satisfying job conditions) and well-being of employees and increasing employee involvement and commitment [18,19]. Furthermore, it has been shown that a good organizational climate has an effect not only on the performance of an organization [20] but on the well-being of employees [21,22]. Provision should also be made for the identification of psychological health issues and to determine the support resources and support systems during extreme situations [2].

Another factor we researched was ensuring the physical safety of workers (provision of security measures, access to testing, flow management, etc.). Ensuring this aspect in workplaces was rated best by the research participants (3.72). Research has showed that the perception of safety and the threat and risk of contagion was one of the biggest perceived stressors during the COVID-19 pandemic [6,23,24]. This should be seen as a priority for the administrations of the organizations, as only an employee who feels physically safe can feel well psychologically.

Another highlighted factor contributing to safety and stability was the administration’s listening and responding in reaction to the difficulties expressed by the staff. The mean of this factor was the lowest (2.72). Therefore, during the extreme events such as those associated with the COVID-19 pandemic, it is essential to take on healthcare workers’ needs, providing timely psychosocial and mental health support, particularly for those groups identified at risk [25]. It is important to create additional conditions for healthcare and pharmacy specialists who are at risk to receive help inside or outside the organization. Healthcare professionals expect to be welcomed, listened to, supported, and protected by their organizations [13]. The absence of this and psychological support during a crisis could cause serious psychological problems for health professionals [26]. Before effective approaches to support healthcare workers can be developed, it is critical to understand their specific sources of expectations related to the organizations. Focusing on addressing those concerns should be the primary focus of support efforts.

Assessments of the participants’ organization care (fair workload, salary distribution, etc.) were also analyzed. In research by G. Zerbini (2020), psychosocial support, as well as the quality of rest during leisure time for healthcare specialists, were identified as important resources, and the better adaptation of infrastructure to COVID-19 in a hospital (e.g., sufficient staffing, team maintenance, consistency of work schedules) was singled out as an expected change [27]. This correlates with our findings, which stress the importance of good organizational factors for protecting the physical and mental well-being of health system workers during the pandemic.

Finally, the information provision timeliness was also included in the work organization factors that allowed well-being to be predicted. This justifies the claim that consistent communication provides security for employees and helps them adapt to changes. This aspect of work organization was rated by the participants (3.63) better than the average. The lack of clear information about the risk may lead to individuals “catastrophizing” and imagining the worst, which exacerbates their anxiety [28]. One of the important strategies for pandemic management is timely communication. Regular, systematic, clear and understandable, timely, open and sincere, uncontroversial, and consistent communication from the administration leads to the better adaptation and well-being of employees and reduces stress [2,5,10].

Identifying the organizational factors effects during this crisis on healthcare professional’s well-being is essential to prepare for other health crises [3]. In terms of protective factors, other studies have emphasized adequate training and the individual perception of societal support to healthcare workers during a pandemic, quality of clinical, managerial and ministerial leadership, and credible measured media coverage [29]. Leaders and managers need to have the appropriate knowledge, skills, and tools to support their staff during these challenging times [30]. The development and maintenance of a good organizational climate is needed to actively promote servant leadership in healthcare settings [31,32,33]. Gigliotti (2016) emphasizes that the crisis leader needs to discharge two roles: engaging in authentic human acts and delivering institutional messages [34].

In summary, according to our research, the management of healthcare and pharmaceutical institutions is advised to take care of the health and well-being of the workers during extreme situations—to maintain the organizational climate and provide physical protection responsibly and as quickly and effectively as possible; to communicate with the employees; introduce organizational solutions focused on employee safety and stability, as well as enhancing bottom-up communication; to listen to the needs of workers, identifying and optimizing the factors that interfere with their safety and stability; to ensure optimal distribution of workloads, salaries and rest time; and to care consistently about timely information and the training of employees.

It is important to discuss several limitations of the study. The study sample consisted of respondents who voluntarily agreed to participate in the study, the majority of whom were females. Thus, future research should consider representing an even wider range of healthcare and pharmacy workers’ experiences and should consider searching for proactive ways to reach an underrepresented sample of male participants. Furthermore, self-reported answers were analyzed, which is inevitably subjective. For future research, it is recommended that repeated measurements are used in order to objectively assess changes in well-being. Moreover, a study conducted from an experimental design would provide more information on the direction, strength, and nature of the relationships analyzed in the article between the changes in physical and psychological well-being and work organization factors that support safety and stability.

It should be noted that the dependence of SAS-SOF statements to the same group was confirmed by factor analysis, internal consistency (Cronbach’s Alpha), and split-half reliability coefficients, which suggests that the cumulative estimate of individual work organization areas may also be a relevant indicative assessment which would provide a better understanding of how to foster employees’ well-being during extreme situations. In further research, it would be meaningful to analyze which specific factors shape employees’ positive or negative perception of separate safety and stability enhancing organizational factors and to deepen the knowledge of the overall indicators of the well-being of healthcare workers. It would allow recommendations to be refined and optimized for the heads of institutions which seek to ensure employee welfare.

## 5. Conclusions

It has been found that the identification of organizational factors and their proper management was significant for the physical and psychological well-being of healthcare workers during the lockdown. Assessments of the analyzed areas of work organization during the pandemic (i.e., organizational psychological climate stability; caring for employee’s health and well-being; ensuring the protection of employees and control of the spread of infection; absence of confrontation between administration and employees; organizational support and assistance to the employee in overcoming difficulties; distribution of workloads and salaries following the principle of justice; timely provision of the necessary information to employees; consistency of the content of communications to employees; as well as their total score) were positively related to the lockdown-related subjective physical and psychological well-being changes of healthcare workers and pharmacy workers.

## Figures and Tables

**Table 1 ijerph-18-08202-t001:** Items of safety and stability supporting organizational factors during pandemic (SAS-SOF) group.

Item’s Marking	SAS-SOF Items	Adapted Framework	Framework; Shanafelt et al. [13]
I	Psychological climate in my organization remained unchanged or even improved.	Psychological needs	Support me
II	My organization showed additional care about employees’ health and well-being (e.g., offered free psychological consultations, provided information on opportunities for emotional support, etc.)	Psychological needs	Support me
III	My organization has adequately ensured employees’ protection and the control of the spread of infection (e.g., provided protective equipment, provided the possibility to have the COVID-19 test, separated staff flows).	Physical protection	Protect me
IV	There was no confrontation between the administration and the staff in my organization.	Response	Hear me
V	The organization’s support and assistance (the administration listening and responding) helped me to overcome difficulties.	Response	Hear me
VI	The distribution of workloads and salaries to all members of the team followed the principle of justice in my organization.	Care	Care for me
VII	My organization timely provided the necessary information to employees.	Timely information	Prepare me
VIII	Messages and instructions given by my organization to the staff did not contradict each other.	Timely information	Prepare me

**Table 2 ijerph-18-08202-t002:** Socio-demographic characteristics of the sample (*N* = 921).

Variable	Total *N* (%)	Physical Well-Being		Psychological Well-Being	
		Decreased *N* (%)	Increased/Has Not Changed *N* (%)	Decreased *N* (%)	Increased/Has Not Changed *N* (%)
Gender					
Women	817 (89.4)	188 (88.7)	605 (89.4)	308 (87.7)	487 (90.2)
Men	97 (10.6)	24 (11.3)	72 (10.6)	43 (12.3)	53 (9.8)
Work field					
Public	652 (71.6)	141 (66.5)	496 (73.5)	239 (68.1)	401 (74.5)
Private	259 (28.4)	71 (33.5)	179 (26.5)	112 (31.9)	137 (25.5)
Specialty					
Physicians	248 (27.1)	56 (26.4)	189 (27.8)	106 (30.1)	140 (25.9)
Nurses	245 (26.7)	54 (25.5)	186 (27.4)	91 (25.9)	150 (27.7)
Pharmacy specialists	224 (24.5)	58 (27.4)	157 (23.1)	93 (26.4)	121 (22.4)
Administrative staff	73 (8.0)	16 (7.5)	53 (7.8)	22 (6.3)	48 (8.9)
Other professions	126 (13.8)	28 (13.2)	94 (13.8)	40 (11.4)	82 (15.2)
Age (years)					
Mean	42.64	39.69	43.43	41.65	43.09
SD	12.72	12.28	12.75	12.82	12.65
Min–Max.	21–79	21–67	21–79	22–79	21–49

**Table 3 ijerph-18-08202-t003:** Characteristics of SAS-SOF indicators and comparison by groups of change of physical and psychological well-being.

SAS-SOF Items Marking	Total Mean (SD) *N* = 921	Physical Well-Being			Psychological Well-Being		
		Decreased *N* = 212	Increased/Has Not Changed *N* = 683	*p* (2-Tailed)	Decreased *N* = 352	Increased/Has Not Changed *N* = 545	*p* (2-Tailed)
		Mean (SD)			Mean (SD)		
I *	2.97 (1.34)	2.37 (1.32)	3.14 (1.30) **	<0.001 **	2.56 (1.30)	3.21 (1.31) **	<0.001 **
II	2.88 (1.49)	2.39 (1.47)	3.00 (1.45) **	<0.001 **	2.53 (1.44)	3.06 (1.47) **	<0.001 **
III	3.72 (1.27)	3.44 (1.29)	3.79 (1.26) **	<0.001 **	3.45 (1.28)	3.87 (1.24) **	<0.001 **
IV	3.29 (1.44)	2.78 (1.46)	3.44 (1.40) **	<0.001 **	2.99 (1.48)	3.46 (1.39) **	<0.001 **
V	2.72 (1.43)	2.25 (1.27)	2.83 (1.44) **	<0.001 **	2.35 (1.29)	2.91 (1.46) **	<0.001 **
VI	3.17 (1.43)	2.68 (1.39)	3.30 (1.41) **	<0.001 **	2.78 (1.35)	3.39 (1.43) **	<0.001 **
VII	3.63 (1.32)	3.24 (1.36)	3.73 (1.29) **	<0.001 **	3.39 (1.33)	3.75 (1.30) **	<0.001 **
VIII	3.51 (1.30)	3.17 (1.35)	3.60 (1.27) **	<0.001 **	3.28 (1.33)	3.64 (1.26) **	<0.001 **
Total estimate	25.89 (8.89)	22.32 (8.52)	26.837 (8.71) **	<0.001 **	23.35 (8.43)	27.30 (8.80) **	<0.001 **

* The SAS-SOF statements group numbers correspond to the numbers of these statements and the corresponding statement formulations in Table 1. ** These variables showed statistical significance.

**Table 4 ijerph-18-08202-t004:** Prediction of changes in physical and psychological well-being during lockdown according to SAS-SOF indicators (binary logistic regression).

SAS-SOF Items		Physical Well-Being Increased/Has Not Changed		Psychological Well-Being Increased/Has Not Changed	
		Univariate	Multivariate **	Univariate	Multivariate **
I *	OR	1.56 ***	1.56 ***	1.45 ***	1.49 ***
	OR 95%	1.38–1.76	1.37–1.78	1.31–1.62	1.33–1.66
	*P*	<0.001 ***	<0.001 ***	<0.001 ***	<0.001 ***
II	OR	1.34 ***	1.32 ***	1.28 ***	1.27 ***
	OR 95%	1.20–1.49	1.18–1.48	1.17–1.41	1.15–1.40
	*P*	<0.001 ***	<0.001 ***	<0.001 ***	<0.001 ***
III	OR	1.23 ***	1.23 ***	1.30 ***	1.30 ***
	OR 95%	1.10–1.39	1.08–1.39	1.17–1.44	1.17–1.46
	*P*	<0.005 ***	<0.005 ***	<0.001 ***	<0.001 ***
IV	OR	1.38 ***	1.41 ***	1.26 ***	1.30 ***
	OR 95%	1.24–1.54	1.25–1.58	1.14–1.38	1.17–1.43
	*P*	<0.001 ***	<0.001 ***	<0.001 ***	<0.001 ***
V	OR	1.35 ***	1.33 ***	1.33 ***	1.33 ***
	OR 95%	1.20–1.52	1.17–1.50	1.20–1.47	1.20–1.48
	*P*	<0.001 ***	<0.001 ***	<0.001 ***	<0.001 ***
VI	OR	1.36 ***	1.33 ***	1.35 ***	1.34 ***
	OR 95%	1.22–1.52	1.19–1.49	1.23–1.49	1.21–1.48
	*P*	<0.001 ***	<0.001 ***	<0.001 ***	<0.001 ***
VII	OR	1.32 ***	1.33 ***	1.23 ***	1.25 ***
	OR 95%	1.18–1.48	1.18–1.51	1.11–1.36	1.12–1.39
	*P*	<0.001 ***	<0.001 ***	<0.001 ***	<0.001 ***
VIII	OR	1.30 ***	1.35 ***	1.23 ***	1.28 ***
	OR 95%	1.15–1.45	1.19–1.54	1.11–1.37	1.15–1.44
	*P*	<0.001 ***	<0.001 ***	<0.001 ***	<0.001 ***
Total estimate	OR	1.06 ***	1.06 ***	1.05 ***	1.06 ***
	OR 95%	1.04–1.08	1.04–1.09	1.04–1.07	1.04–1.08
	*P*	<0.001 ***	<0.001 ***	<0.001 ***	<0.001 ***

* The SAS-SOF statements group numbers correspond to the numbers of these statements and the corresponding statement formulations in Table 1. ** Controlled indicators in multivariate analyses—gender, age, workplace, position. *** These variables showed statistical significance.

## Data Availability

All databases are available from the corresponding author upon reasonable request.

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
