# Peer review of "How Does the Assessment of Work Organization during the COVID-19 Pandemic Relate to Changes in the Well-Being of Health System Workers?"

_ijerph, 2021, doi:10.3390/ijerph18158202_

Round 1

Reviewer 1 Report

Thank you for the opportunity to review this timely article regarding how changes in healthcare and pharmacy specialists' physical and psychological well-being during the first wave of COVID-19 lockdown are related to work organization factors. The paper reflects a significant and meaningful topic to examine these changes. There are minor concerns that should be taken into account before considering this article for publication.

  • It would be better to add the study design in the title
  • It is suggested to mention the period of 'the first wave of COVID-19' in the abstract and the article.
  • In lines 75-80, please clarify whether the study assessed physical and psychological well-being or only psychological well-being? It's a bit unclear for 'this article analyses part of the data collected.'
  • In lines 87-89, ‘222 target healthcare institutions and pharmacy chains' were selected. What is the sampling method? How to choose the healthcare institutions and or any inclusion criteria? Or all the healthcare institutions and pharmacy chains were invited?
  • In lines 91-94, the respondents were the employees of the healthcare institutions and pharmacy chains. I want to clarify whether the administrative staff had a healthcare background or were the healthcare specialists as the term 'healthcare and pharmacy specialists' were used throughout the paper. Do you mean healthcare workers/staff as 'health system workers' is used in your title? Would you please state clearly about the 'representatives of other professions? Who are they?
  • Sessions 2.1.3 and 2.2.1 repeated paragraphs of Safety And stability supporting organizational factors (lines 101-115 & lines 117-131).
  • Table 2 (line 208) shows that the numbers of Physical well-being and Psychological well-being do not match the total numbers. Were there missing answers except those mentioned in line 144?
  • In line 254, when did the first lockdown happen?
  • Any limitations of the study? Any potential sources of bias identified? Please consider adding, if any.

Thank you!

Author Response

Thank you very much for your comments and remarks!

  • It would be better to add the study design in the title. Due to diverse reviewers recommendations after extensive discussions we decided to maintain the title as it is.
  • It is suggested to mention the period of 'the first wave of COVID-19' in the abstract and the article. The period of the first lockdown has been mentioned several times according to your comment.
  • In lines 75-80, please clarify whether the study assessed physical and psychological well-being or only psychological well-being? It's a bit unclear for 'this article analyses part of the data collected.' It has been clarified.
  • In lines 87-89, ‘222 target healthcare institutions and pharmacy chains' were selected. What is the sampling method? How to choose the healthcare institutions and or any inclusion criteria? Or all the healthcare institutions and pharmacy chains were invited? It has been clarified.
  • In lines 91-94, the respondents were the employees of the healthcare institutions and pharmacy chains. I want to clarify whether the administrative staff had a healthcare background or were the healthcare specialists as the term 'healthcare and pharmacy specialists' were used throughout the paper. Do you mean healthcare workers/staff as 'health system workers' is used in your title? Would you please state clearly about the 'representatives of other professions? Who are they? It has been clarified.
  • Sessions 2.1.3 and 2.1 repeated paragraphs of Safety And stability supporting organizational factors (lines 101-115 & lines 117-131). It has been corrected.
  • Table 2 (line 208) shows that the numbers of Physical well-being and Psychological well-being do not match the total numbers. Were there missing answers except those mentioned in line 144? Table No. 2 (line 208) presents the distributions of all subjects (N = 921 - the number of subjects indicated in the table title) according to the research questions. As different numbers of subjects did not provide data on their different characteristics (eg gender, work field, specialty, age, physical well-being, psychological well-being), Table 2 shows the number of subjects by columns Physical well-being and Psychological well-being do not match the total numbers.

Part of the missing answers in the SAS-SOF question group was restored according to the Person mean imputation procedure (Newman, 2014) - this procedure is mentioned in lines 142-146. Reconstructed SAS-SOF data are used for statistical calculations (data presented in Tables 3 and 4).

In response to the remark, adjustments were made in the text (line 143-144) explaining in which tables (results presented in Table 3 and Table 4) the data after imputation are applied. The misused and misleading insert (N = 921) was also deleted.

  • In line 254, when did the first lockdown happen? It has been clarified.
  • Any limitations of the study? Any potential sources of bias identified? Please consider adding, if any. More information has been added.

Thank you!

Reviewer 2 Report

The work deals with the important issue of creating employee-friendly working conditions in times of crisis - in this case, the COVID-19 pandemic. The numerous works available in the public domain prove that the well-being of health workers during a pandemic, with increased workload, is critical to patient safety. A discouraged, inefficient, tired employee becomes dangerous for the patient over time. The presented study deals with this aspect.
The research tools and methods used in the work should be assessed as appropriate, adequate for the time in which the research was conducted and reliable. The CAWI method is one of the most frequently used methods of collecting scientific data during the forced reduction of contacts between people during the COVID-19 pandemic and therefore should be considered appropriate. Noteworthy is a large group of research participants, as well as the authors' care for a wide representation of professional groups. The reviewer finds it valuable to compare the results in various breakdowns (gender, ownership of the institution, profession).
The results obtained in Table 3 concerning the influence of individual factors on the work environment are particularly valuable. It should be emphasized in the assessment that the authors pay attention to both the physical and psychological aspects of working conditions, which, unfortunately, many employers do not pay attention to.
The discussion conducted in the work does not raise doubts in the assessment, the authors correctly refer to the collected literature data and correctly confront them with the results obtained by them.
The reviewer, both in terms of the content, methodology and research, does not make any critical comments. recommends the publication of the work in its current form.

Author Response

Thank you very much for your valuable comments and suggestions!

Reviewer 3 Report

This is an interesting study in which a large sample is used to delve into an aspect related to COVID-19 and worker well-being. A series of recommendations follow:

TITLE

This could be maintained, although it is slightly long. It is recommended that a shorter option be considered, although if this is not possible because it would eliminate information, it could be left as it is. I think it is a good idea to start the manuscript with a question as it has been done, since it captures the reader's attention.

ABSTRACT

It provides a lot of information and is structured. 

In terms of considerations, the words "Introduction", "Methods", "Results", and "Conclusions" could be eliminated.

The keywords are well selected. Perhaps they could be put in alphabetical order if they all have the same relevance. As for the study design, it is recommended that one of the keywords refer to it. 

INTRODUCTION

The information presented is accurate but scarce. More theoretical background is needed. In recent months, much information has been generated that can be located in databases such as SCOPUS or Web of Science. 

An objective is stated in the last paragraph. It is recommended, given the theoretical framework, to propose a hypothesis associated with it. 

MATERIALS AND METHODS

The information presented is adequate, but it is recommended that it be included in other subsections to be in line with other articles and Journals, and thus facilitate the reader's search for the information, using a known scheme. It is recommended that the contents of this section be restructured so that it has the following subsections:

  • Procedure (or Design and Procedure). This subsection can describe the design followed, the procedure, how the information was gathered, etc.
  • Participants. Description of the main sociodemographic data concerning the participants.
  • Instruments.
  • Data analysis.

The section on ethical aspects of the original manuscript can be maintained using the same subtitle or included in the Procedure section.

RESULTS

Adequate. The format of the tables, which is very neat, is noteworthy. 

The information provided is useful and relevant.

DISCUSSION

The data are discussed and further elaborated upon. However, it would be necessary to link this part to a greater extent with the theoretical framework of the introduction. In the discussion, more of the bibliographic references used in the introduction should be used. In addition, as mentioned at the beginning, the theoretical framework should be expanded as it is relevant but brief. 

More emphasis should be placed on limitations and future lines of research. 

CONCLUSIONS

Adequate.

OTHER ASPECTS

With respect to "Author Contributions", acronyms should be used instead of the full name and surname. To this end, on the first page, after the first and last names, the acronyms should be placed, and these acronyms would be used to describe the contribution of each member of the group. 

In summary, this is an article that analyzes a hot topic that presents the results correctly and contributes to further study of the phenomenon under study. 

Author Response

Thank you very much for your comments and remarks! Please see the attachment.
